# Theory-of-Mind Enhanced Dialogue Generation in Situated Contexts

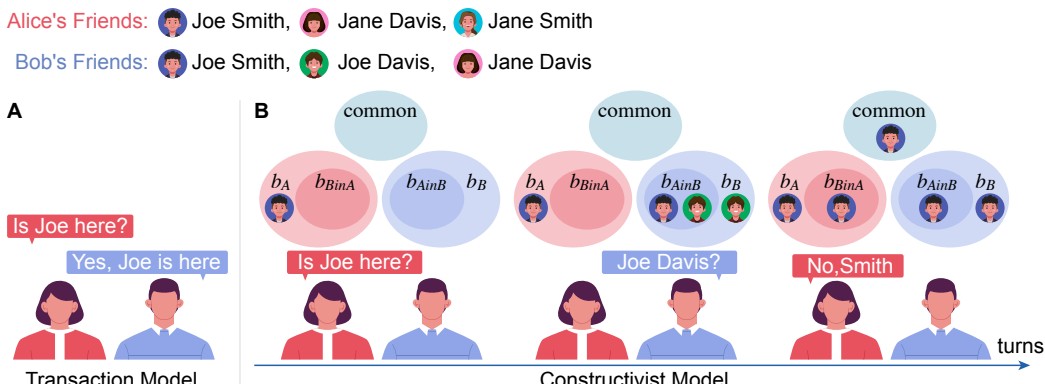

Figure 1: **Transaction Model vs Constructivist Model**. According to the Transaction Model, when Alice (in red) asks about "Joe", Bob will respond based on his individual knowledge, assuming there is no ambiguity in Alice's questions. In contrast, from the perspective of the Constructivist Model, Bob will negotiate the meaning of "Joe" until both confirm the same person.

## Abstract

Humans talk in free-form while negotiating the expressed meanings or common ground. Despite the impressive conversational abilities of the large generative language models, they do not consider the individual differences in contextual understanding in a shared situated environment. In this work, we propose MindDial, a novel conversational framework that can generate situated free-form responses to negotiate common ground. We design an explicit mind module that can track three-level beliefs – the speaker's belief, the speaker's prediction of the listener's belief, and the common belief based on the gap between the first two. Then the speaking act classification head will decide to continue to talk, end this turn, or take task-related action. We augment a common ground alignment dataset MutualFriend (He et al., 2017b) with belief dynamics annotation, of which the goal is to find a single mutual friend based on the free chat between two agents. Experiments show that our model with mental state modeling can resemble human responses when aligning common ground meanwhile mimic the natural human conversation flow. The ablation study further validates the third-level common belief can aggregate information of the first and second-order beliefs and align common ground more efficiently.

## 1 Introduction

Humans communicate based on a shared understanding of contextual meanings (also known as negotiated meanings or common ground) (Burleson, 2007; Delia & O'Keefe, 1982). Though significant improvements have been achieved in the natural language generation community with large generative language models (Radford et al., 2019; Brown et al., 2020; Raffel et al., 2020; Ouyang et al., 2022) for providing *general*, *helpful* and *objective* information based on users' queries (Glaese et al., 2022; OpenAI, 2023), modeling common ground in daily chit-chats that deliver *situated*, *free-form* and *subjective* responses is rarely explored. Critically, two vital components are not carefully addressed in previous dialogue models:

**I. Shared Situated Context.** Conventional neural dialogue agents model dialogues as multi-turn question answering, where they treat conversational agents as sender and receiver: the sender initiates the topic by sending out a query and the receiver takes in the query and produces the next sentence with the highest probability by learning from a massive dialogue corpus. These models are considered *Transaction models*. Refer to Figure 1A for an example, Bob responds according to his individual knowledge and assumes there is no ambiguity in Alice's question. However, these models differ from the nature of human communication — we communicate based on a shared comprehension of contextual meanings. Such perspective is particularly essential when the dialogue agents can only partially perceive the environment and effective communication can only occur when they negotiate to obtain a common ground. These models are in line with the *Constructivist model*. As shown in the talking pairs from Figure 1B, Alice and Bob have to negotiate to confirm the common understanding of "`Joe`" is "`Joe Smith`".

**II. Individual Mental Dynamics.** One prerequisite for the Constructivist framework is that agents have to explicitly model individual differences in understanding, intention and goals, *i.e.*, mental states. Compared with the Transaction framework that only models the individual's state, a proper inference of the conversational partner's state can result in faster convergence of the common ground. Moreover, based on the individual mental state modeling, one can easily produce free-form dialogues, *i.e.*, the goal is not to be forced to generate a single response based on the context but to keep speaking or stop based on individual differences.

In order to step towards real-world situated conversation, in this work, we introduce **MindDial**, a new dialogue framework built upon the theory-of-mind (ToM) modeling in cognitive theory, aiming at modeling free-form neural dialogue generation with the Constructivist model's point of view. It is worth noting that some works simulate human values through their feedback to enable dialogue systems to generate responses aligning better with human expectations (Bai et al., 2022; Yuan et al., 2022). Different variants of Sally-Anne test are also proposed to test ToM of large language models (Kosinski, 2023; Ullman, 2023; Sileo & Lernould, 2023). In comparison, we consider mind modeling in situated daily dialogue scenarios and differing from value alignment and ToM question answering from three perspectives:

- **A structured "five mind" representation** (Fan et al., 2021): as the Constructivist model example shown in Figure 1B, we model two first-order beliefs, two second-order beliefs of each other's mind, and the third-level common belief;
- **Belief dynamics prediction**: we explicitly model how each utterance in the context results in the occurrence, disappearance, or no change of the entities in the agents' beliefs as the belief dynamics. Then, the final belief distribution will be the summation of all belief dynamics estimation over utterances in the dialogue history. It avoids the problem of losing track of some entities if we directly model the final belief given a long dialogue input;
- **Common belief modeling**: we define the common belief as how probably the current speaker thinks about an entity that will be aligned to the common ground. We hypothesize that the common belief is based on the gap between the speaker's belief of the physical world and her belief estimation of the listener.

**Contribution** We consider our contributions as five-fold: i) We propose a novel framework for free-form dialogue generation from the cognitive perspective, which integrates dialogue generation, planning, and mind predictions. ii) We design an explicit mind module that can track the first and second-order beliefs over long contexts by aggregating belief dynamics. A third-level common belief based on the gap between the two will support the next response generation. iii) We augment a collaborative dialogue dataset MutualFriend with belief dynamics annotations for each utterance that can serve as a new benchmark dataset for ToM in situated dialogue tasks. iv) The evaluation results and the ablation analysis demonstrate the efficacy of each component in our mind module. Human evaluation shows that the responses generated with the three-level beliefs are more accurate and efficient for negotiating common ground. v) We provide preliminary results for large language models indicating that though they can estimate the belief changes in the dialogue but cannot apply the estimation for dialogue response generation.

## 2 RELATED WORK

**Theory-of-Mind (ToM)** ToM is a crucial capability for human social interactions developed in early life (Kovács et al., 2010; Richardson et al., 2018). In literature, early works model belief update through time in sequential games with partially observable Markov decision process (POMDP) (Baker et al., 2011; De Weerd et al., 2013; Doshi et al., 2010; Han & Gmytrasiewicz, 2018). One agent's belief update is based on the estimate of others' current beliefs, resulting in an infinite recursion. However, in real life, studies have shown that humans could go no deeper than two levels of recursion (Camerer et al., 2004). Therefore, works (Fan et al., 2021) began the efforts to end the recursion when their beliefs merge into the "common mind".

Modeling the belief of others has been extensively studied in symbolic-like environments (Wunder et al., 2011; Rabinowitz et al., 2018; Kleiman-Weiner et al., 2016; Ho et al., 2016), where agents need to incorporate or compete for a goal. Efforts to measure models' ability to recognize false beliefs and perspective-taking also emerge in robotics (Yuan et al., 2020; Milliez et al., 2014), computer vision (Eysenbach et al., 2016; Fan et al., 2021), and natural language processing (Qiu et al., 2022; Nematzadeh et al., 2018) using the Sally-Anne test (Baron-Cohen et al., 1985). It is also shown that augmenting the model with external mind modules can help improve the performance of tasks involving intensive belief exchange and rich social interaction scenarios (Fan et al., 2021; Qiu et al., 2022). In this work, we explore how the common belief modeling with first and second-order belief difference can enhance the quality and efficiency of the response generation in dialogue tasks.

**Neural Dialogue Generation** Neural dialogue generation has made impressive progress after various datasets and advanced model architectures are proposed. Both SEQ2SEQ and decoder-based models (Lewis et al., 2019; Zhang et al., 2020) are introduced into the open-domain dialogue systems for style and personality-controlled generation (Hu et al., 2022; Cho et al., 2022), with knowledge and emotion-aware abilities (Varshney et al., 2022; Liu et al., 2022), etc. In addition, researchers explore reinforcement learning-based methods to enable agents to learn from human feedback (Bai et al., 2022), coordinate, and compete with each other in task-oriented dialogues (Verma et al., 2022; Jang et al., 2022).

**Cooperative Communication** For a cooperative task, efficient communication could be essential, especially in a situation when each agent can only have a partial observation of the environment. To guarantee that the communication takes the least cost meanwhile provides the most informative messages, previous works proposed multiple methods to align the common ground between agents (Bohn et al., 2019; Anderson, 2021). Specially for dialogue tasks, datasets have been collected to provide golden utterances when people try to align the common ground with each other based on structured knowledge (He et al., 2017a), in situated tasks (Bara et al., 2021; Kim et al., 2019), multimodal and continuous environment (Haber et al., 2019; Udagawa & Aizawa, 2021). Frameworks have been adopted to model the belief dynamics using GNN, RNN, and transformers (He et al., 2017a; Udagawa & Aizawa, 2021; Qiu et al., 2022). Fan et al. (2021) introduce low-level visual cues that may possibly indicate mind transition. However, most of the models only focus on the first-order belief (the current speaker's belief of the world). In this work, we track the speaker's both first and second-order beliefs (the current speaker's belief of others) and demonstrate how the jointly modeling between the two can help align the common ground.

## 3 THE MINDDIAL FRAMEWORK

The dialogue corpus can be denoted as $\mathcal{D} = \{(U_n, KB_n^p, E_n^p, C_n, Y_n)\}_{n=1}^{N}$, where $U_n = (u_{n,1}, ..., u_{n,K})$ represents the dialogue history and $K$ is the number of turns. $KB_n^p = (kb_{n,1}, ...kb_{n,I})$ where $I$ is the number of knowledge passages. $p \in A, B$ represents the two agents. We assume that the current speaker is A, and $p$ will be dropped for the following formulation. The knowledge base contains entities under different attributes. $E_i$ is a union set of entities visible for the current speaker. Each entity in $E_n = \{E_n^U, E_n^{KB}\}$ is either shown in the dialogue history or in the speaker's knowledge base. $Y_n = \{y_n^1, .., y_n^L\}$ is A's next response consisting of several utterances. $C_n = \{c_n^1, ..., c_n^L\}$ is A's corresponding speaking act sequence, with $c_n^1, ..., c_n^{L-1}$ is "continue to talk" and $c_n^L$ belongs to stop talking or make task related action. Given the dialogue contexts and a partial response $\{y_n^1, .., y_n^{l-1}\}$, speaker A will first decide the next speaking act $c_l$. If she decides to talk, the next utterance $y_n^l$ will be generated.

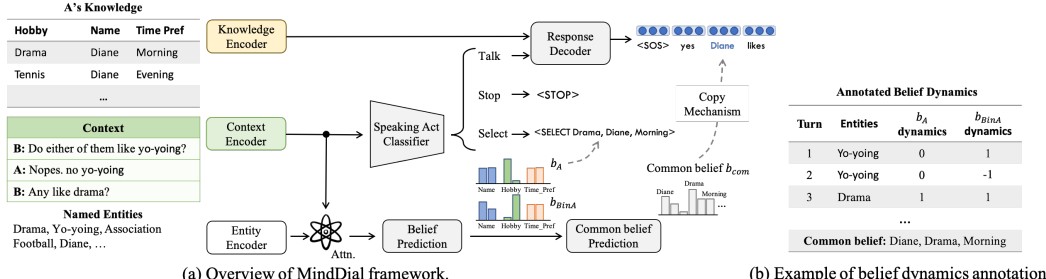

Figure 2: (a) Illustration of the MindDial framework. The colored barplots denote the belief prediction's outputs $b_A$ and $b_{BinA}$ for each attribute; the barplot in lightgray denotes the normalized probability of the next entity to be mentioned w.r.t. the common belief prediction module. (b) We annotate belief dynamics of each turn for the current context; refer to Section 4.1 for details.

**Utterance Encoder** The utterance and the structured knowledge encoder are built upon sequential models like recurrent neural networks (Cho et al., 2014) or a transformer encoder (Vaswani et al., 2017). We take the last hidden state as the turn-level representation $s_{u_k}, s_{kb_i} \in \mathbb{R}^{1 \times d_h}$. The sentence-level representation is $s_U$ and $s_{KB}$. When we flatten the turns and passages into a single sequence, the encoded output is written as $o_U$ and $o_{KB}$. The entity representation $s_E$ is a linear transformation of the corresponding word embedding of the entities.

**Belief prediction** We define the first and second-order belief as the confidence distribution over entities of each attribute. For a knowledge base with $M$ attributes, the belief is $b = \{b^m = (b^{m_1}, ..., b^{m_{J_m}})\}_{m=1}^M, \sum_j b^{m_j} = 1$. $J_m$ is the number of entities for attribute $m$. We further define belief dynamics as the state change of each entity at each time step $k$ as $\Delta b_k$ with each entry value ranging from $-1$ to $1$. $-1$ indicates the disappearance of an entity in the belief and $1$ indicates an occurrence of an entity. Then the current belief is the accumulated prediction of the dynamics over all dialogue turns: $b^m = \text{softmax}(b_0^m + \sum_k \Delta b_k^m)$. $b_0$ is initialized to all zeros. The belief dynamics of speaker $A$ are obtained by calculating the attention score between the current utterance and the entity shown in her own knowledge base

$$\Delta b_{k,A}^m = tanh(s_{E^{KB}}^m \cdot (s_{u_k})^T), \qquad d_A^m = \sum_k (b_{k,A}^m)^T \cdot s_{E^{KB}}^m, \qquad (1)$$

where $s_{E^{KB}}^m \in \mathbb{R}^{J_m \times d_z}$ concatenates all entity representations for the attribute $m$ shown in $E^{KB}$. $d_A^m \in \mathbb{R}^{1 \times d_z}$ is the belief representation of attribute $m$ for the speaker. Since A's current estimate of B's knowledge base is based on her own knowledge base along with the entities mentioned by B in the context, using the same attention module applied to the context and the whole entity set of the given attribute $s_E^m$, we can get $d_{BinA}, b_{BinA}$:

$$\Delta b_{k,BinA}^m = tanh(s_E^m \cdot (s_{u_k})^T), \qquad d_{BinA}^m = \sum_k (\Delta b_{k,BinA}^m)^T \cdot s_E^m. \qquad (2)$$

We further define common belief $b_{com}$ as how likely each entity is agents' next talking focus over all possible entities based on the gap $bdiff$ between $b_A$ and $b_{BinA}$. The output range is between 0 and 1. 1 indicates that this entity will be mentioned in the next response and 0 otherwise. The common belief will be learned through another attention layer $b_{com} = \sigma(s_E \cdot d_{bdiff}^T)$, where $s_E \in \mathbb{R}^{|Ent| \times d_z}$. $|Ent| = \sum_m J_m$ is the total number of entities over all attributes.

To get the gap representation $d_{bdiff}$, we first aggregate $d_A$ and $d_{BinA}$ over entities by a weighted summation of $p^m$: $d_{bdiff}^m = (p^m d_A^m + (1-p^m)d_{BinA}^m)$, where $p^m = \sigma(W_{ent}[b_A^m, b_{BinA}^m]^T)$, and $W_{ent} = \mathbb{R}^{1 \times 2d_z}$. $d_{bdiff}^m$ is then aggregated over attributes: $d_{bdiff} = g(W_{Att}[d_{bdiff}^0; ...; d_{bdiff}^M])$, where $W_{Att} = \text{softmax}(W_0, ..., W_M)$. $W_m$ is computed by the Jensen–Shannon divergence (Lin, 1991) over $b_A^m$ and $b_{BinA}^m$ in that we hope to pay more attention to attributes if they differ a lot between $b_A$ and $b_{BinA}$. $g$ is a linear transformation.

**Speaking act Classifier**   We divide the speaking act into 3 categories: {continue to talk, end the current turn, take task-related action}. Based on the current context and the partial response, the action is predicted using $p_{c^l} = \text{MLP}(s_U)$, where $\text{MLP}(\cdot)$ denotes a multi-layer perceptron network.

**Response Decoder**   We take another recurrent neural network or a transformer decoder as our response generator. For each word prediction, it receives the embedding vector $y_{t-1}$ of the word predicted at time-step $t-1$ and outputs the last hidden state $h_t \in \mathbb{R}^{1 \times d_{z'}}$ and $P_{vocab}(w_t)$ over the fixed vocabulary obtained from the training set.

*Multi-source Copy Mechanism.* We follow Bai et al. (2021); He et al. (2017a) to adopt the copy mechanism so that the final word distribution depends on both the decoder output and a copy probability of words shown in dialogue history, speaker's knowledge base and the common belief. The context representation of utterance and structured knowledge at each time step is obtained through the Attention module:

$$d_t^\phi = \text{softmax}(h_t \cdot o_\phi^T)o_\phi \tag{3}$$

where $\phi \in \{U, KB\}$. Then the decoder state $h_t$ attends over the dialogue history representation $d_t^U$ and the knowledge representation $d_t^{KB}$ by

$$\alpha_t = \text{softmax}(h_t \cdot [d_t^U; d_t^{KB}]^T), \qquad d_t = \alpha_t[d_t^U; d_t^{KB}] \tag{4}$$

where $\alpha_t = (\alpha_t^U, \alpha_t^{KB}) \in \mathbb{R}^{1 \times 2}$ is used to combine the distributions of the two inputs as shown in Equation (5). We also use a generation probability $p_t^{gen} \in [0, 1]$ to balance the distribution between input sources and the fixed vocabulary, where $p_t^{gen} = \sigma(W_{gen}[y_{t-1}, h_t, d_t]^T)$, and $W_{gen} \in \mathbb{R}^{1 \times (d_{emb} + d_z + d_{z'})}$. Besides, we set a mind weight $p_{com}$ to leverage the common belief distribution into the final prediction. The overall distribution is obtained by

$$P(w_t) = (1 - p_{com})\left[p_t^{gen}P_{vocab}(w_t) + (1 - p_t^{gen})\sum_{\phi:\phi\{U,KB\}}\alpha_t^\phi P_\phi(w_t)\right] + p_{com}\text{softmax}(b_{com}) \tag{5}$$

**Objective**   Mean squared error (MSE) loss will be used to measure the difference between the predicted and ground truth belief dynamics. The common belief prediction loss will be measured by Binary Cross Entropy (BCE). The action classification head is updated by Cross-Entropy loss. Apart from belief and act training loss, we use the NLL loss to capture the word order information:

$$\mathcal{L}_{NLL} = -\frac{1}{|y^l|}\sum_{t=1}^{|y^l|}\log(P(y_t^l|y_{1:t-1}^l, U, KB)) \tag{6}$$

The final loss is composed of three parts:

$$\mathcal{L} = \mathcal{L}_{NLL} + \mathcal{L}_{belief} + \mathcal{L}_{act} \tag{7}$$

## 4 EXPERIMENTS

### 4.1 SETTINGS

**Dataset**   To provide a reasonable quantitative measure of belief dynamics in the dialogue, the expected dataset should contain rich belief exchanges. Meanwhile, the belief exchange and the final common ground can be easily labeled. In the absence of appropriate datasets, we augment MutualFriend (He et al., 2017b) with belief annotation to evaluate our dialogue generation framework for its clear definition of belief (distribution over structured knowledge) and common ground (the mutual friend). In the MutualFriend task, each agent has a private knowledge base including a list of friends and their attributes like name, school, *etc*. There is a shared friend that both agents have and they need to chat with each other to find this mutual friend. We only keep the successful dialogues and the final data split for train/val/test is 4922/608/581. Each dialogue in the training set contains a maximum of 53 turns and each turn with a maximum length of 29.

To get the supervision signal for belief dynamics, we manually label each entity after one turn of utterance as occur (mentioned by the speaker), no change (not mentioned in the last turn), or disappear (negated by the speaker). Figure 2 illustrate one annotation process. For example, when B is asking

about "yo-yoing", this entity is marked as 1 for $b_{BinA}$ dynamics. However, since it does not belong to A's knowledge, for the first-order belief of speaker A, we annotate it as no change. Then, when "yo-yoing" is negated by A, it will be marked as a "disappear" in $b_{BinA}$ dynamics. One entity is labeled as the common belief to be aligned next if it is shown in the response utterance.

**Implementation** To serve as a baseline in this task, the model is expected to encode current contexts and predict the belief dynamics. Then it will further generate the next response based on both the dialogue history and the belief prediction. Therefore, we select dialogue baselines from three categories: 1) We use the Gated Recurrent Unit (GRU) (Cho et al., 2014) among the recurrent neural networks for its memory efficiency of modeling sequential data; 2) We combine the powerful encoder Transformer (Vaswani et al., 2017) with the decoder Transformer (Radford et al., 2019) for its strong conversation abilities; 3) We employ pre-trained encoder-decoder Transformer architectures such as BART (Lewis et al., 2020) which can be flexibly adopted to sequence-to-sequence tasks.

For all transformer models, we finetune the pretrained model on the MuturalFriend dataset. For context encoding, we prepend the BOS token at the beginning of the context and use its corresponding hidden representation as the turn and sentence-level representation for the following attention layers and speaking act prediction. The entity encoding will be a linear transformation of the corresponding word embedding. Meanwhile, the decoder's predicted vocabulary distribution will be mediated by the copy mechanism (Bai et al., 2021). The model is trained on a single A6000 GPU for 30k steps with an initial learning rate of 1e-4. The batch size is set to 32. Results are gathered over 3 random seeds. More data format and implementation details can be found in Appendix G and Appendix I.

## 4.2 EVALUATION AND RESULTS

**Mind prediction** We first determine whether the models can accurately track both the first and second-order beliefs. The dynamics prediction performance is evaluated using the macro-average of Precision, Recall, and F1-score. We can see from Table 1 that all three types of encoders can predict the belief dynamics in mind $A$ and $BinA$ fairly well compared with the random guess (0.33).

Table 1: **Belief dynamics prediction**. $\Delta b$ specifies belief dynamics for $b_A$ and $b_{BinA}$.

| Models | $\Delta b$ | Precision | Recall | F1 |
|---|---|---|---|---|
| GRU | | 69.00±0.02 | 83.33±0.02 | 74.67±0.02 |
| Transformer | $b_A$ | 65.67±0.01 | 70.67±0.09 | 67.67±0.04 |
| BART | | 70.67±0.02 | 62.33±0.07 | 64.67±0.03 |
| GRU | | 73.33±0.02 | 83.33±0.02 | 77.33±0.01 |
| Transformer | $b_{BinA}$ | 70.00±0.01 | 69.33±0.11 | 68.33±0.06 |
| BART | | 73.33±0.02 | 60.00±0.07 | 62.33±0.04 |

For the next common entity prediction, we compare the performance using both information from $b_A$ and $b_{BinA}$ as shown in Section 3 with a method computing $d_{bdiff}$ by $d_A$ or $d_{BinA}$ only. In $d_A$ only method, $p^m$ will be set to 1 and the Jensen-Shannon divergence is set to 0 for all attributes. In $d_{BinA}$, $p^m$ will be 0. We also report the same metrics as the belief dynamics. However, since the common belief label is pretty sparse (selecting one/two entities from over 20 entities), all models get similar results. Therefore, we further treat it as a ranking task and use MRR (Mean Reciprocal Rank) to measure how well the target entity can be returned among all available entities. As shown in the first three columns of Table 4, combining $b_A$ and $b_{BinA}$ can achieve higher ranking score, which suggests that people consider both their self belief and the belief estimation of others when choosing the next entity to align with.

**Next Response Generation** We evaluate the response generation by both speaking act prediction accuracy and commonly used textual generation metrics (BLEU, METEOR, and ROUGE (Papineni et al., 2002; Lin, 2004; Lavie & Agarwal, 2007)). For the textual evaluation metrics, we align the generated texts with the ground truth utterances only when both speaking acts are "continue to talk". If the ground truth action is "continue to talk" while the predicted action is "end the current turn", the score will be set to 0. To study the contributions of our mind module, we compare the performance of our full model with generations not copying from common belief distribution ($p_{com} = 0$ in Equation (5)).

From the results shown in Table 2, we can see that generators combined with the external mind module achieve better performances for all three categories. Without considering the mind prediction, the performances drop for both response generation and speaking act prediction. This indicates that

Table 2: **Next response generation and speaking act classification**. For each model category, we report the textual response performance (METEOR, ROUGE-L, BLEU) and action classification accuracy (Action acc). +mind indicates the generator copies from common belief distribution.

| Models | METEOR | ROUGE-L | BLEU-1 | BLEU-2 | Action acc |
|---|---|---|---|---|---|
| GRU | 7.64±0.33 | 9.26±0.90 | 12.65±1.06 | 4.70±0.42 | 76.49±0.48 |
| GRU+mind | 8.56±0.03 | 9.89±0.27 | 12.70±0.56 | 5.17±0.11 | 77.64±0.54 |
| Transformer | 9.93±0.70 | 12.44±0.15 | 13.45±0.21 | 3.95±0.21 | 77.24±0.50 |
| Transformer+mind | 10.45±0.11 | 13.14±0.23 | 14.15±0.07 | 4.90±0.14 | 77.66±0.42 |
| BART | 10.70±0.28 | 12.54±1.90 | 15.10±1.13 | 4.75±0.35 | 75.77±2.62 |
| BART+mind | 11.72±0.26 | 14.01±1.10 | 16.95±1.34 | 6.25±1.06 | 76.90±0.69 |

Figure 3: **Responses generated by model variants**. For each case, the current contexts are shown in the top-right; bottom-right visualizes the ground truth responses and the responses generated by models without mind (w/o mind), with common belief estimation using the first-order belief ($b_A$ only), the second-order belief ($b_{BinA}$ only) and with both ($b_A + b_{BinA}$).

reasoning about the belief dynamics can help the model resemble human responses when aligning common ground meanwhile form human-like speaking flows.

**Case Study** Figure 3 demonstrates how the response differs with and without mind/first/second-order belief modeling in Transformer category. The example on the left shows that the model without belief modeling can falsely pull the conversation towards "Julia". Model $b_A$ and $b_{BinA}$ respond with entities in their corresponding order of beliefs but do not address the A's false belief of "Julia" and "Randy". For the right example, though all models can capture the correct knowledge and generate reasonable responses, only $b_A+b_{BinA}$ provides additional information of other unknown attributes. More examples can be found in Appendix B and Appendix C.

### 4.3 Ablation study

To assess the contributions of each component in the proposed method, we derive the following variants as an ablation study:

- *No dynamics*: We predict the current beliefs $b_A$ and $b_{BinA}$ directly from the given $K$ turns of dialogues instead of summing over belief dynamics across turns in contexts.
- $b_A$ *only* / $b_{BinA}$ *only*: The common belief distribution is learned only based on the first or second-order beliefs (the same in Section 4.2).

**Will the belief prediction with dynamics be more accurate?** We contrast belief prediction with and without belief dynamics in the GRU category. We can observe from Table 3 that predictions by summing belief dynamics over turns can reach higher precision and F1. We reckon that modeling the state change of entities per utterance can help better track beliefs in longer contexts.

**Does each order of belief benefit the response generation?** From the pair-wise comparison between +$b_A$ only and +$b_{BinA}$ of columns 4-5 in Table 4, we can see that +$b_A$ performs better in BART while +$b_{BinA}$ is better in the other two models. And jointly modeling both two levels of beliefs can achieve the best results. This suggests that there is no clear dominance between the first and second-order beliefs. Our full model $b_A+b_{BinA}$ can weight the information of the two beliefs accordingly based on the current given contexts.

Table 3: **Belief estimation with/without dynamics prediction and Human study**. w/o $\Delta b$ represents models trained directly with $b_A$ and $b_{BinA}$. w/ $\Delta b$ denotes models trained with belief dynamics. Comparisons are made between our Full model vs. $b_A$ only vs. nomind model. The Full vs. GT is a comparison between the full model and referenced ground truth.

| | | Belief estimation | | Human study | | | |
|---|---|---|---|---|---|---|---|
| Models | Belief | Precision | F1 | Pairs | Wins. | Lose. | Tie. |
| w/o $\Delta b$ | $b_A$ | 45.00±0.00 | 46.00±0.00 | Full vs. $b_A$ only | 46.30 | 43.20 | 10.50 |
| | $b_{BinA}$ | 45.33±0.01 | 46.33±0.01 | Full vs. w/o mind | 46.30 | 10.50 | 43.20 |
| w/ $\Delta b$ | $b_A$ | 53.67±0.01 | 56.00±0.01 | Full vs. GT | 47.20 | 52.80 | - |
| | $b_{BinA}$ | 55.00±0.00 | 57.00±0.01 | | | | |

Table 4: **Common belief prediction, next response generation and self-talk simulation by ablating the first and second-order belief**. Rows with model+$b_A$ specify results with common belief predicted by the first-order belief only, with model+$b_{BinA}$ by the second-order belief only and with +$b_A$+$b_{BinA}$ by both the first and second-order beliefs.

| Models | F1 | MRR | ROUGE-L | Action acc | Success rate | # of Turn | # of Entity |
|---|---|---|---|---|---|---|---|
| GRU+$b_A$ | 60.00±0.00 | 26.75±0.31 | 8.89±1.12 | 77.54±0.66 | 6.58±1.25 | 37.74±0.13 | 12.80±2.90 |
| +$b_{BinA}$ | 60.00±0.00 | 26.65±0.51 | 9.88±0.26 | 77.63±0.55 | 6.33±1.01 | 37.77±0.25 | 12.78±2.86 |
| +$b_A$+$b_{BinA}$ | 60.00±0.00 | 26.87±0.43 | 9.89±0.27 | 77.64±0.54 | 6.92±1.60 | 37.71±0.17 | 12.78±2.94 |
| Transformer+$b_A$ | 58.67±0.01 | 24.70±1.74 | 12.63±0.09 | 77.43±0.04 | 9.99±4.82 | 19.20±0.94 | 12.23±2.72 |
| +$b_{BinA}$ | 59.00±0.01 | 25.12±1.77 | 12.75±0.55 | 77.77±0.11 | 8.83±2.18 | 17.79±1.28 | 10.55±0.99 |
| +$b_A$+$b_{BinA}$ | 58.67±0.01 | 25.17±2.06 | 13.14±0.23 | 77.66±0.42 | 10.76±4.55 | 17.65±0.29 | 10.91±2.73 |
| BART+$b_A$ | 60.00±0.00 | 26.46±0.71 | 14.41±1.29 | 76.84±0.55 | 5.77±0.04 | 31.77±0.21 | 15.31±2.84 |
| +$b_{BinA}$ | 59.67±0.01 | 25.93±1.03 | 12.83±1.43 | 76.25±1.58 | 7.37±3.00 | 28.93±1.84 | 13.71±4.92 |
| +$b_A$+$b_{BinA}$ | 60.00±0.00 | 26.51±0.65 | 14.10±1.10 | 76.90±0.69 | 7.65±2.60 | 30.68±4.57 | 14.87±4.52 |

**Common ground alignment**    We further use self-talk to evaluate how well the agents align their common ground with and without modeling the gap between their beliefs and the belief of others. For each round, the model will play as both the speaker and the listener to generate the next response. The game will end when the two agents exceed the maximum of turns, or both of the agents decide to make the choice. When the model predicts the next action is "take task-related action", we will choose the friend with the maximum probability based on $b_A$ as its choice of friend. Since models without a mind cannot make this choice, we only compare models with $b_A$ or $b_{BinA}$ only and our full model. We report the average success rate when the agents choose the same person in their knowledge bases as their mutual friend at the end of each game round. Meanwhile, we also calculate the average number of turns they use to end the game and the average number of entities they mention when they reach the common ground.

The right three columns of Table 4 show the self-talk results. Compared with $b_A$ and $b_{BinA}$ only, our full model can achieve higher success while take less turns in the game and mention fewer number of entities to achieve the alignment. This indicates that attending both the first and second-order beliefs helps the model align the common ground more efficiently.

**Human Evaluation**    We ask 20 well-educated students to run human studies on 1,000 examples. A pair-wise comparison is made between our Full model and models with first-order beliefs, models without mind modules, referenced ground truth, considering the grammar correctness and efficiency of finding the common friend. From Table 3, we can observe that our model generates more grammarly correct and efficient responses and also more resembles human responses. Please refer to Appendix F for the interface details.

### 4.4    HOW WELL DO THE LARGE LANGUAGE MODELS (LLMS) PERFORM THIS TASK?

LLMs have gained great attention for their impressive conversational abilities. Instead of directly asking ChatGPT questions regarding beliefs and false beliefs like the Sally-Anne test (Kosinski, 2023; Ullman, 2023; Sileo & Lernould, 2023), we investigate its capability of solving cooperative communication tasks involving intense belief exchanges. As shown in Figure 4, we initialize two ChatGPT models as two chat agents. Instructions are given to both of them about the goal and rules

Table 5: Belief estimation

| $\Delta$ b | Precision | F1 |
|---|---|---|
| $b_A$ | 65.00 | 53.87 |
| $b_{BinA}$ | 71.00 | 55.83 |

Table 6: Common belief prediction, next response generation and self-talk simulation.

| F1 | MRR | ROUGE-L | Action Acc | Success | # of Turn | # of Entity |
|---|---|---|---|---|---|---|
| 63.00 | 30.31 | 6.48 | 43.79 | 2.33 | 40 | 20.21 |

of the game. Additional prompts are given to help circulate the conversation meanwhile probe their belief estimation and action prediction (see Appendix E). From the belief estimation and generation results in Table 5 and Table 6, we can observe that ChatGPT can predict the belief change fairly well when directly querying it. However, the low performance on the generation results indicates that it talks without taking the belief estimation into account. We further investigate this problem by looking into the generation details.

As marked in Figure 4, we observe several potential questions of current LLMs: 1) **Inappropriate belief estimation:** when ChatGPT1 proposes friends interested in L&L Hawaiian Barbecue and outdoor activities, ChatGPT2's first-order belief over possible entities is also outdoor which is not consistent with his knowledge base; 2) **Mind inconsistent utterance**: ChatGPT1 is asking Apache Software Foundation while she believes more in people working for American Motors Corporation as the mutual friend; 3) **No belief tracking:** when the two agents confirm with L&L Hawaiian barbecue and outdoor, ChatGPT1 returns back to Pelican Products which it already negated before. Please check more examples in Appendix H.

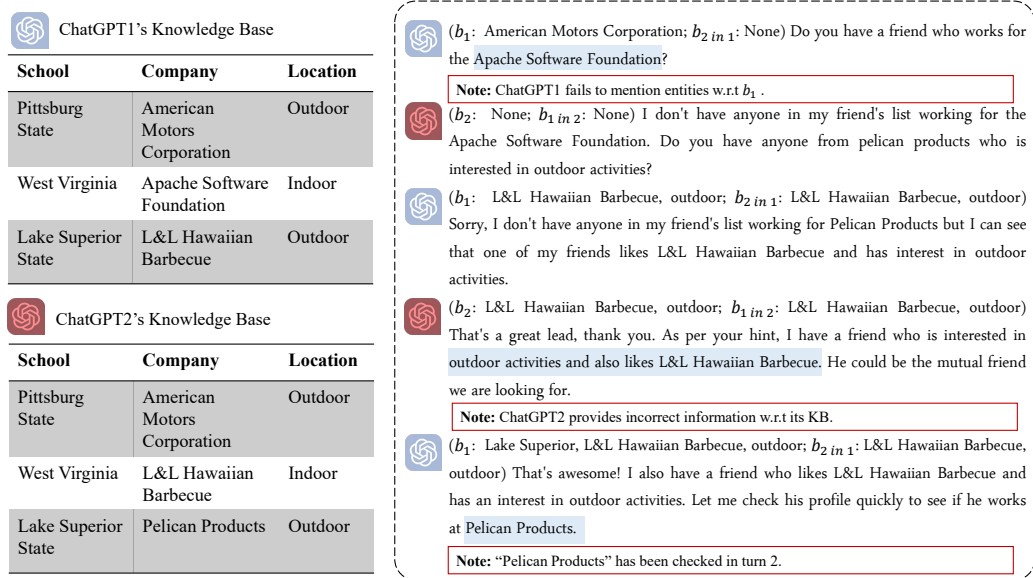

Figure 4: **Self-talk simulation using ChatGPT**. The left table contains the knowledge bases of two ChatGPT agents. The right box shows the conversation flow. We additionally prompt ChatGPT to obtain their belief estimation as shown in parenthesis. Refer to Supplementary Material for detailed prompt templates.

## 5   CONCLUSION

In this study, we present MindDial, a novel framework for generating human-like dialogues. Our approach incorporates an external mind module, which predicts the first and second-order beliefs of the speaker. The response generation takes into account a third-level common belief, which is determined based on the disparity between the first two levels. Through extensive experiments, we demonstrate that responses that consider belief estimation can enhance common ground negotiation between agents. Our ablation studies further validates the effectiveness of our design in capturing belief dynamics and modeling common beliefs by aggregating the first and second-order beliefs.

## ETHICS STATEMENT

Generating dialogues while considering Theory-of-mind modeling can offer advantages to social chatbot applications by facilitating seamless and effective communication and collaboration between robots and humans. For instance, when a person mentions "medicine," the robot can promptly provide the appropriate medication to address an urgent situation. On the flip side, if chatbots exhibit a more human-like demeanor, there is the potential for negative social consequences, such as making people addicted to social media. It's important to note, however, that our proposed methods primarily focus on introducing scientific concepts and investigations and do not aim for such potentially problematic applications. Furthermore, our work strictly relies on publicly available benchmarks and data, ensuring that no personal information is collected. Our human study is conducted with utmost care to safeguard participants' privacy and interests, with no potential harm involved.

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

# Appendix

We provide supplementary materials as follows:

- Appendix B: Examples of self-talk simutaion of our full model.
- Appendix C: Examples contrasting self-talk results of the full model with the first or second-order only models.
- Appendix D: Response generation results when the response generator copies from the common mind with different copy weights.
- Appendix E and Appendix H: Prompts and examples for ChatGPT self-talk simulation.
- Appendix F: Human evaluation interface.
- Appendix G: Hyperparameters for model training.
- Appendix I: Dataset annotation format

## A  LIMITATIONS

It's important to acknowledge that our framework relies on a structured knowledge dataset that allows for straightforward annotation of belief dynamics. This limits its application in more diverse scenarios involving open-ended dialogues, where the mind annotation may become ambiguous. Additionally, the self-talk simulation may exhibit repetitive questions and similar expression patterns, potentially stemming from overfitting on a small dataset. Despite these limitations, we hope our framework can serve as a starting point for future endeavors aimed at developing conversational agents that can truly emulate human-like conversations.

## B  SELF-TALK SIMULATION - FULL MODEL

In this section, we demonstrate three examples of the self-talk simulation with our full model for Section 4.3 Common ground alignment. Before each utterance, we also provide the agents' beliefs of the most probable mutual friend in parentheses. In Figure 5 and Figure 6, we can see that both agents can mention relevant entities (in bold) in consecutive turns and the common ground converges effectively (A's belief of the most probable mutual friend equals B's belief of the most probable mutual friend). In the third example, B can promptly negate the wrong entity. Also, when A mentions "afternoon", B can timely update her belief meanwhile spot this unique attribute in her table and provide relevant entity "dog sport".

## C  SELF-TALK SIMULATION - ABLATION

We compare the self-talk results ablating both the first and second-order belief in our model. For the first example in Figure 8, with the full model, when B mentions "Atmospheric Sciences & Meteorology", A can respond by providing relevant information. However, model with $b_A$ only fails to capture mutual entities and keeps listing all possible majors and companies. Model with $b_{BinA}$ only brings up more entities than necessary as shown in the last two utterances. For the second example in Figure 9, combing both first and second-order beliefs can help correct false beliefs about "Electrical Engineering" meanwhile gradually talking towards the common ground. In comparison, the model with $b_A$ only uses more turns to hit the matched entities while $b_{BinA}$ only provide false information thus miss the correct entity. As for the third example in Figure 10, models considering the second-order beliefs ($b_A + b_{BinA}$ and $b_{BinA}$ only) can find mutual entities efficiently. Model with $b_A$ only may take more turns since it can mismatch to other entities without timely false belief correction.

## D  HOW MIND PREDICTION INFLUENCES THE GENERATION RESULTS

We investigate the effects of changing the copy weight of the common belief distribution when generating the next response. The line plot of Figure 11 gives an illustration of the variation of the

**A's Knowledge Base**

| Hobby | Major | Location |
|---|---|---|
| Rowing | European Studies | Outdoor |
| Gardening | Electrical Technology | Outdoor |
| Beach Volleyball | Latin | indoor |
| Gardening | European Studies | indoor |
| Gardening | Religious Education | indoor |
| Weightlifting | European Studies | Outdoor |
| Beach Volleyball | Humanities | Outdoor |
| Rowing | Afro-American Studies | Outdoor |
| Rowing | Protestant Affiliation | Outdoor |

**B's Knowledge Base**

| Hobby | Major | Location |
|---|---|---|
| Weightlifting | History | indoor |
| Martial Arts | Religious Education | indoor |
| Gardening | Religious Education | indoor |
| Weightlifting | Science Technologies | indoor |
| Freestyle Football | Religious Education | indoor |
| Sculling | Electrical Technology | indoor |
| Sculling | Islamic Studies | indoor |
| Movie Memorabilia Collecting | Humanities | indoor |
| Baton Twirling | Russian | indoor |

A: (Gardening, Religious education, Indoor)
Hi

B: (Martial Arts, Religious education, Indoor)
Hi

A: (Gardening, Electrical Technology, Outdoor)
I have 2 friends who like Outdoor

B: (Sculling, Electrical Technology, Indoor)
I have 3 friends who like the Indoor , do you ?

A: (Gardening, Electrical Technology, Outdoor)
I have 1 friend who into **Gardening**

B: (Gardening, Religious Education, Indoor)
I have a friend who works at **Religious Education**

A: (Gardening, Religious Education, Indoor)

Figure 5: **Full-model example 1**. B can find the corresponding entity "Religious Education" under Major when A mentions "Gardening".

**A's Knowledge Base**

| School | Time | Name | Location |
|---|---|---|---|
| University of Texas at Brownsville | Evening | Kyle | Outdoor |
| University of Texas at Brownsville | Evening | Linda | Outdoor |
| Youngstown State University | Evening | Cheryl | Indoor |
| Youngstown State University | Evening | Linda | Outdoor |
| Texas A&M International University | Morning | Linda | Outdoor |

**B's Knowledge Base**

| School | Time | Name | Location |
|---|---|---|---|
| Youngstown State University | Evening | Cheryl | Indoor |
| State Univ of New York at Binghamton | Morning | Cheryl | Indoor |
| University of Texas at Brownsville | Evening | Cheryl | Indoor |
| University of Texas at Brownsville | Morning | Janice | Indoor |
| Texas A&M International University | Evening | Janice | Indoor |

A: (Texas A&M International University, Morning, Linda, Outdoor):
Hi

B: (State Univ of New York at Binghamton, Morning, Cheryl, Indoor):
Hi

A: (Youngstown State University, Evening, Linda, Outdoor):
I have 2 friends who like Evening , like Outdoor

B: (State Univ of New York at Binghamton, Morning, Cheryl, Indoor):
I have 2 friends who like the Morning , like Indoor

A: (Youngstown State University, Evening, Linda, Outdoor
I have 2 friends who like the **Youngstown State University**

B: (Youngstown State University, Evening, Cheryl, Indoor):
I have 2 friends who like the **Cheryl**

A: (Youngstown State University, Evening, Cheryl, Indoor)

Figure 6: **Full-model example 2**. B can respond with the correct name "Cheryl" after A provides an entity of School.

**A's Knowledge Base**

| Hobby | Major | Time |
|---|---|---|
| Dog Sport | Legal Assisting | Afternoon |
| Rock Climbing | Legal Assisting | Morning |
| Rock Climbing | Ministry & Church Administration | Morning |
| Rock Climbing | Computer Engineering | Morning |
| Rock Climbing | Forestry | Afternoon |
| Swimming | Multi Studies | Morning |
| Drawing | Chemistry | Morning |
| Rock Climbing | Chemistry | Afternoon |
| Skateboarding | Veterinary Sciences | Afternoon |

**B's Knowledge Base**

| Hobby | Major | Time |
|---|---|---|
| Squash | Multi Studies | Evening |
| Cryptography | Chemical Engineering | Evening |
| Cryptography | Conservation Mgmt. & Protection | Evening |
| Dog Sport | Geophysical Engineering | Evening |
| Dog Sport | Legal Assisting | Afternoon |
| Cryptography | Special Education | Evening |
| Dog Sport | Legal Assisting | Evening |
| Dog Sport | Psychology | Evening |
| Dog Sport | Forestry | Evening |

A: (Skateboarding, Veterinary Sciences, Afternoon)
Hi

B: (Dog Sport, Legal Assisting, Afternoon)
Hi

A: (Rock Climbing, Legal Assisting, Morning)
I have 2 friends who like Morning

B: (Dog Sport, Legal Assisting, Evening)
I have 2 friends who like the Evening

A: (Rock climbing, Legal Assisting, Morning)
I have 2 friends who like the **Rock Climbing**

B: (Dog Sport, Legal Assisting, Evening)
**I have none of those**

A: (Rock climbing, Legal Assisting, Morning)
I have 2 friends who like the **Afternoon**

B: (Dog Sport, Legal Assisting, Afternoon)
I have 2 friends who like **Dog Sport**

A: (Dog Sport, Legal Assisting, Afternoon)

Figure 7: **Full-model example 3**. B can timely correct A's false belief of "Rock Climbing". Also, B can update her belief to entities with "Afternoon" corresponding to A's last utterance.

**A's Knowledge Base**

| Major | Company | Time |
|---|---|---|
| Atmospheric Sciences & Meteorology | Ocean Spray | Morning |
| Construction & Building Tech. | Monster Beverage Corporation | Morning |
| Adult & Continuing Education | Hyland Software | Morning |
| Social Science Education | Sonic Restaurants, Inc | Morning |
| Business Education | Nautilus, Inc | Afternoon |

**B's Knowledge Base**

| Major | Company | Time |
|---|---|---|
| Business Education | Underwriters Laboratories | Afternoon |
| Social Science Education | Sonic Restaurants, Inc | Afternoon |
| Science Technologies | Moog Incorporated | Afternoon |
| Business Education | Kelly Services | Afternoon |
| Atmospheric Sciences & Meteorology | Ocean Spray | Morning |

$b_A + b_{BinA}$:
A: I have 2 friends who like morning
B: I have 3 friends who like the afternoon
A: I have 1 friend who likes the Sonic Restaurants, Inc
B: I have a friend who likes the **Atmospheric Sciences & Meteorology**
A: I have a friend who likes the **Ocean Spray**
− − − − − − − − − − − − − − − − − − − − − − −
$b_A$ only:
A: I have a friend who works at Sonic Restaurants, Inc.
B: I have a friend that works at Business Education
A: I have a friend who works at Atmospheric Sciences & Meteorology
B: I have a friend who works at Moog Incorporated
A: I have a friend who works at Monster Beverage Corporation
……
− − − − − − − − − − − − − − − − − − − − − − −
$b_{BinA}$ only:
A: All my friends like the morning
B: All of my friends like the afternoon except 1
A: Do any of your friends work at Sonic Restaurants, Inc. ?
B: I have a friend who works at Business Education who likes the Ocean Spray who works at Atmospheric Sciences & Meteorology
A: I have a friend who works at Nautilus, Inc. who works at Monster Beverage Corporation

Figure 8: **Ablation example 1**. The model with $b_A + b_{BinA}$ can use less turns and fewer entities to find the mutual friend.

**A's Knowledge Base**

| Hobby | School | Major |
|---|---|---|
| Netball | Marietta College | Equestrian Studies |
| Graffiti | University of the District of Columbia | Museums & Preservation |
| Sculling | Saint Joseph's College | Middle Eastern Studies |
| Netball | Saint Mary's College | Ecology Sciences |
| Slot Car Racing | Saint Joseph's College | Biology |
| Slot Car Racing | Marietta College | Special Education |

**B's Knowledge Base**

| Hobby | School | Major |
|---|---|---|
| Slot Car Racing | Marietta College | Special Education |
| Letterboxing | Eastern New Mexico University | Special Education |
| Rock Climbing | Portland State University | Electrical Engineering |
| Slot Car Racing | Washington State University | Biology |
| Road Biking | Michigan State University | Electrical Engineering |
| Slot Car Racing | Eastern New Mexico University | Electrical Engineering |

**$b_A + b_{BinA}$:**
A: I have 2 friends who like the Slot Car Racing
B: I have 2 friends who like **Electrical Engineering**
A: **I have none of those**
B: I have 2 friends who like Special Education
A: I have a friend who works at Marietta College

- - - - - - - - - - - - - - - - - - - - - - - - -

**$b_A$ only:**
A: Do you have any friends who like Netball ?
B: I have a friend that works at Electrical Engineering
A: I have a friend that works at Marietta College
B: I have a friend that works at Slot Car Racing
A: I have a friend that works at Saint Joseph's College
B: I have a friend that works at Special Education

- - - - - - - - - - - - - - - - - - - - - - - - -

**$b_{BinA}$ only:**
A: I have 3 friends who work at Slot Car Racing
B: I have none of those
A: I have a friend who works at Saint Joseph's College
B: I have a friend who works at Portland State University
A: I don't have any friends who work there
......

Figure 9: **Ablation example 2**. The model with $b_{BinA}$ only sometime provide false negation information (in the second turn) thus take longer time to find the mutual friend.

**A's Knowledge Base**

| Major | School | Name |
|---|---|---|
| Legal Assisting | Judson College | Thomas |
| Range Management | University of Detroit Mercy | Jane |
| Mathematics | Villa Julie College | Keith |
| Mathematics | Saint Joseph's College | Jordan |
| Range Management | Pacific Union College | Alice |
| Art, Music, & Drama Edu | Saint Joseph's College | Gabriel |
| Art, Music, & Drama Edu | University of New Mexico | Howard |
| Range Management | Judson College | Theresa |
| Art, Music, & Drama Edu | Ursuline College | Jordan |

**B's Knowledge Base**

| Major | School | Name |
|---|---|---|
| Range Management | University of Detroit Mercy | Jane |
| Electrical Engineering | Purdue University North Central | Gabriel |
| Aviation & Air Transportation | University of Detroit Mercy | Thomas |
| Computer Programming | Saint Cloud State University | Harold |
| Electrical Engineering | Saint Cloud State University | Gabriel |
| Mining & Mineral Engineering | University of Detroit Mercy | Thomas |
| Architectural Engineering | University of Detroit Mercy | Jacqueline |
| Architectural Engineering | Saint Cloud State University | Keith |
| Range Management | Saint Cloud State University | Harold |

**$b_A + b_{BinA}$:**
A: I have 3 friends who like the **Range Management**
B: I have 2 friends who like **Saint Cloud State University**
A: **I have none of those**
B: I have 2 friends who like **University of Detroit Mercy**
A: I have a friend who works there

- - - - - - - - - - - - - - - - - - - - - - - - -

**$b_A$ only:**
A: Do you have any friends who like Range Management ?
B: I have a friend that works at Saint Cloud State University
A: I have a friend that works at Art, Music, & Drama Edu
B: I have a friend that works at Gabriel
A: I have a friend that works at saint joseph's college
......

- - - - - - - - - - - - - - - - - - - - - - - - -

**$b_{BinA}$ only:**
A: I have 3 friends who work at Art, Music, & Drama Edu
B: I have none of those
A: I have a friend who works at Range Management
B: I have 2 friends who work at University of Detroit Mercy

Figure 10: **Ablation example 3**. The model with $b_A$ only cannot timely correct the false beliefs so as to fail to capture the mutual entity.

performance when $p_{com}$ in $\{0, 0.2, 0.4, 0.6, 0.8\}$. We observe that the generation score first increases and then decreases as the weights vary. This indicates that the model not only needs to learn what should be mentioned next but also how to express it in the response. It is supposed to trade-off between entity name prediction accuracy and grammar fluency.

## E   CHATGPT PROMPT

```
→ (At the beginning of the first turn): I will introduce a game to
  you called "mutual friend". There will be two players in the game.
  Each of the player will have one table. Each row of the table is one
  friend the player has. The column is the attributes of the friends
  such as the company they work for, their hobbies and their location
  preference for leisure activities. Through chat, these two players
  need to find out who is the mutual friend between them. Please note
  the players cannot observe the tables of the other. They need to
  provide information in their table or ask information from other's
  table. The goal of the game is to find out the mutual friends using
  less turns and less entities in the table. I'm playing the game with
  [Alice|Bob]. Here is my table (the knowledge base of [Bob|Alice]):

  hobby, name, location

  Surfing, Jane, Outdoor

  ...

  (After each turn):

→ [Alice|Bob] said: {last generated response}. What should I say next?
  Please provide one sentence as the response from my perspective:

→ Which row do you think is the most probable mutual friend in my table?

→ Based on the information in my table and the current conversation,
  which entity do you think Bob believes could be the mutual friend?

→ Based on the current context, do you think I should make selection now
  or chat more with [Alice|Bob]?
```

Figure 12: Template for ChatGPT self-talk prompt.

## F   HUMAN STUDY INTRODUCTION INTERFACE

Figure 13 shows our interface for instructing the human subjects to compare the generation results of different models.

## G   IMPLEMENTATION DETAILS

We adopt both the Recurrent Neural Network and transformers as our backbones:

**Gated Recurrent Unit (GRU)** GRU belongs to the Recurrent Neural Network family with a gating mechanism. It is capable of maintaining an internal state of the sequential information with faster computation. We use one GRU as the encoder and another as the decoder.

# Mutual Friend Task

Here is a game called 'Mutual Friend'. There will be two players in the game. Each of the players will have one table. Each row of the table is one friend the player has. The column is the attributes of the friends such as the company they work for, their hobbies, and their location preference for leisure activities. Through chat, these two players need to find out who is the mutual friend between them. Please note the players cannot observe the tables of the other. They need to provide information at their table or ask for information from other's tables through chat. The game's goal is to find mutual friends using fewer turns and fewer entities in the table.

Here is an example:

Agent 0 and Agent 1 have their own private tables shown below

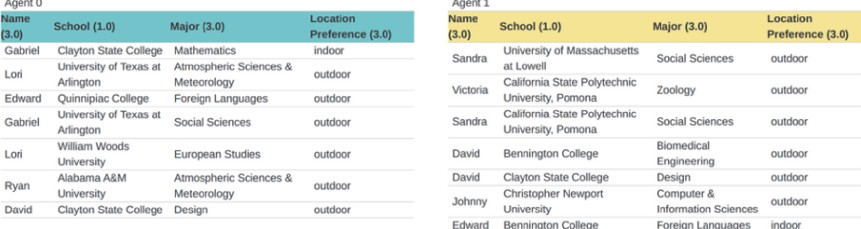

Through chat, they exchange information and finally find out their mutual friend.

| Agent 1 (human) | Okay so, all outdoors but one |
|---|---|
| Agent 0 (human) | I have one outdoor friend |
| Agent 1 (human) | Johnnny? |
| Agent 0 (human) | Gabriel is my indoor |
| Agent 0 (human) | I have 2 Loris |
| Agent 1 (human) | I have two davids |
| Agent 0 (human) | I have one david that went to Clayton State |
| Agent 1 (human) | SELECT (Clayton State College \|\| David \|\| Design \|\| outdoor) |
| Agent 0 (human) | SELECT (Clayton State College \|\| David \|\| Design \|\| outdoor) |

In this survey, you will be given a dialogue history between two speakers <speaker0> and <speaker1>, and the private knowledge of the current speaker. Then, you will evaluate three sentences based on grammar correctness, efficiency, and similarity to the ground truth sentence if they are considered as the next dialogue response.

**Grammar correctness:** which of the three sentences is more grammarly correct?

**Efficiency:** which of the three sentences expresses the information more efficiently by faster converging the talk meanwhile mentioning fewer entities?

Agent 0's knowledge:                                    Dialogue History:

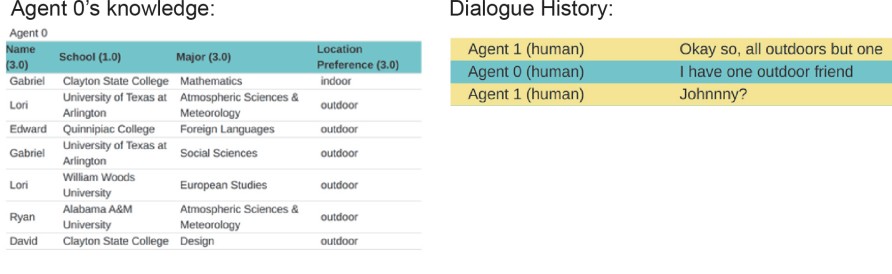

Sentence 1: No Johnny
Sentence 2: Gabriel?
Sentence 3: No, do you have a David?

Figure 13: Human Study Introduction Interface

**Transformers** We employ the BERT-base-uncased[1] as our encoder for context encoding and belief prediction. Then the DistilGPT2[2] will mediate the predicted word distribution with the copy mecha-

---

[1] https://huggingface.co/bert-base-uncased
[2] https://huggingface.co/distilgpt2

nism[3] to generate the final response. Meanwhile, we also directly finetune the encoder-decoder model BART-base[4] in our task.

Table 7 shows the backbones and hyperparameters we use to train our model. We also provide the special tokens BOS, EOS and End Turn used in the corpus. BOS marks the beginning of a dialogue. EOS denotes the end of one utterance. End Turn represents when the agent ends its current turn. For different backones, the special tokens can be different (shown in the right three columns).

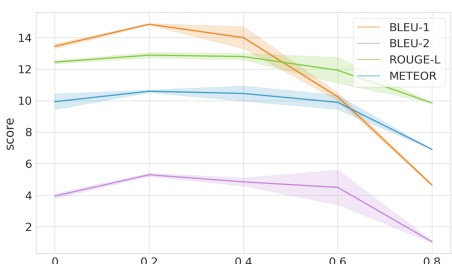

Figure 11: The textual generation score with different copy weights of $p_{com}$.

Table 7: **Hyperparameters for each model category**

| Category | Encoder | Decoder | Hidden size | Batch size | Epochs | BOS | EOS | End Turn |
|---|---|---|---|---|---|---|---|---|
| GRU | GRU | GRU | 200 | 256 | 20 | \<s\> | \</s\> | \<stop\> |
| Transformer | BERT-base | GPT2-distilled | 768 | 32 | 4 | [CLS] | [SEP] | \</s\> |
| BART | BART-base | | 768 | 32 | 4 | \<s\> | \</s\> | \<stop\> |

## H  SELF-TALK SIMULATION - CHATGPT

In this section, we provide additional examples of the self-talk simulation using two ChatGPTs. As the questions we observed in Section 4.4, they fail to talk about cooperative or relevant entities. As shown in Figure 14, when ChatGPT1 mentions "Washington State", ChatGPT2 ignores it and proposes a new school "Randolph-Macon". In the second example (Figure 15), after agreeing with ChatGPT1, ChatGPT2 is expected to provide other attributes along with "freestyle football" for efficiency. There are other cases when the agent can provide false information (Figure 16).

## I  DATA FORMAT

Figure 17 shows the dataset structure with the input context, knowledge tables, next response, speaking act label, and the belief annotations.

---

[3]https://github.com/jq2276/Learning2Copy
[4]https://huggingface.co/facebook/bart-base

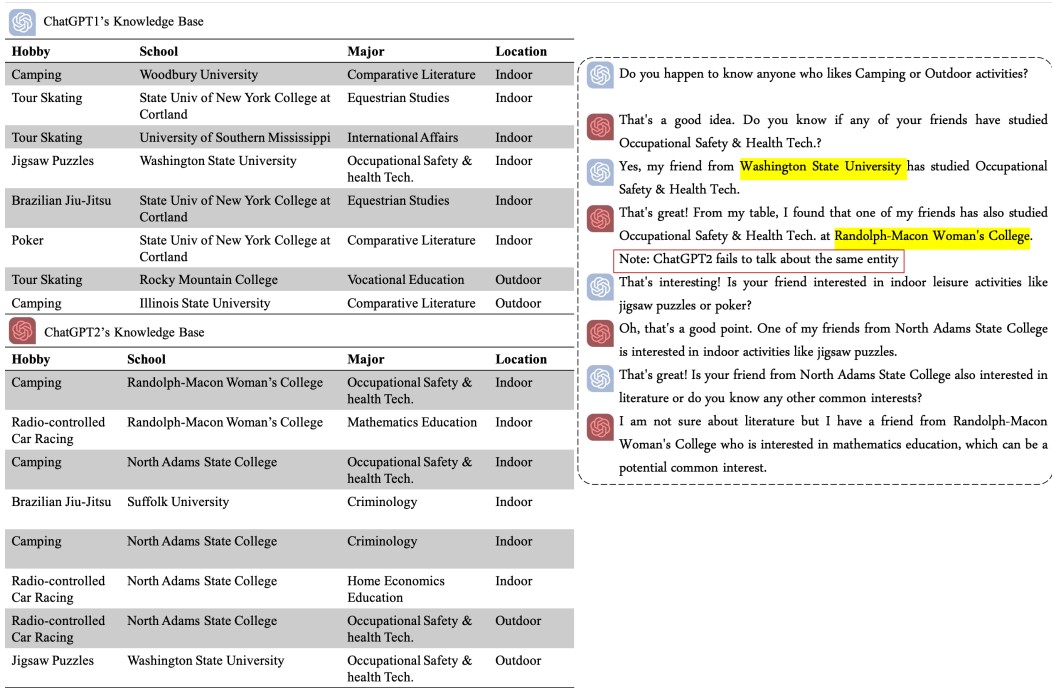

Figure 14: **ChatGPT example 1**. ChatGPT2 diverges the conversation from "Washington State University".

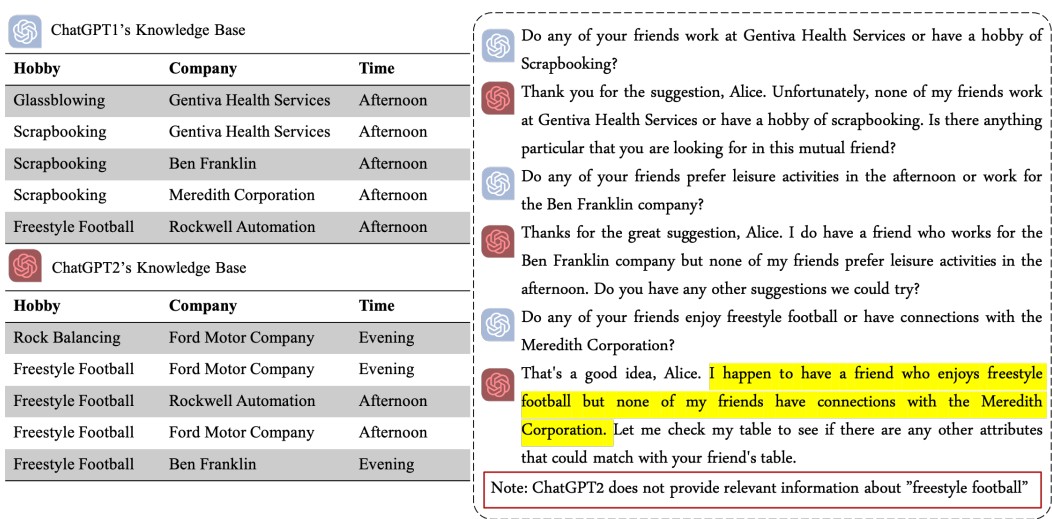

Figure 15: **ChatGPT example 2**. ChatGPT2 is expected to provide more relevant entities with "freestyle football".

ChatGPT1's Knowledge Base

| Hobby | Name | Time |
|---|---|---|
| Mahjong | Juan | Afternoon |
| Gunsmithing | Denise | Afternoon |
| Web Surfing | Denise | Afternoon |
| Wood Carving | Ruth | Evening |
| Drama | Diane | Morning |
| Mineral Collecting | Diane | Afternoon |
| Kitesurfing | Madison | Afternoon |
| Bowling | Juan | Afternoon |
| Flying | Madison | Afternoon |
| Yo-yoing | Diane | Evening |
| Web Surfing | Diane | Afternoon |

ChatGPT2's Knowledge Base

| Hobby | Name | Time |
|---|---|---|
| Cryptography | Julia | Morning |
| Association Football | Harry | Evening |
| Bowling | Juan | Morning |
| Dancing | Janice | Morning |
| Water Sports | Andrea | Evening |
| Action Figure | Diane | Morning |
| Cryptography | Janice | Morning |
| Bowling | Donald | Morning |
| Kitesurfing | Julia | Morning |
| Drama | Diane | Morning |
| Wood Carving | Diane | Morning |

Hey Bob, do you have any friends who are interested in mineral collecting or wood carving?

Well, I do have a friend who enjoys wood carving. Her name is Diane. What other hobbies or interests do you have on your table?

Oh, that's great to hear! I also have Diane on my table and she is interested in drama and yo-yoing. Do you have any other friend who shares the same hobbies as her?

I don't have any friends who share the same hobbies as Diane, but I have a friend named Julia who enjoys water sports and kitesurfing. Do you have any friends with those hobbies on your table?

Note: ChatGPT2 falsely negates the entity

That's interesting! I also have a friend named Madison who is interested in kitesurfing and flying. Could she be the mutual friend we are looking for?

Unfortunately, Madison is not the mutual friend we are looking for, since none of my friends share those interests. Do any of your friends have a hobby or interest in cryptography or association football?

I see. None of my friends share those hobbies, but I do have a friend named Denise who is interested in gunsmithing and web surfing. Does any of your friends share these interests with her?

No, sorry. None of my friends have those interests. Let me check my table again. Maybe we missed something. Are there any other hobbies or interests on your table that we haven't discussed yet?

Figure 16: **ChatGPT example 3**. ChatGPT2 falsely negates that Diane likes drama.

```
{
    'dialogs':[
        {
            'knowledge':[
                [
                    ['hobby', 'name', 'location']
                    ['surfing', 'jane', 'outdoor']
                    ...
                ],
                [
                    ['hobby', 'name', 'location']
                    ['car racing', 'Dan', 'outdoor']
                    ...
                ],
            ],
            'turns':[
                [
                    ['Hi'],
                    ['I have a friend named Jane']
                ],
                [
                    ['I have a Jane too']
                ],
                ...
            ],
            'dynamics':[
                {
                    'utterance': ['I have a friend named Jane'],
                    'hobby': [
                        [0, 0, 0, 0, 0], <bA dynamics>
                        [0, 0, 0, 0, 0], <bBinA dynamics>
                    ]
                    'name':[
                        [0, 0, 0, 1, 0], <bA dynamics>
                        [0, 0, 0, 0, 0], <bBinA dynamics>
                    ]
                    ...
                }
                ...
            ],
            'response': 'Does she like surfing?'
            'action': 'continue to talk'
            'common':{
                    'hobby': [0, 1, 0, 0, 0],
                    'name': [0, 0, 0, 0, 0],
                    ...
            }
        }
    ]
}
```

Figure 17: **Data format**.

