# OpenReview forum: "Theory-of-Mind Enhanced Dialogue Generation in Situated Contexts"
_ICLR.cc/2024/Conference — ICLR 2024 Conference Withdrawn Submission_

### Official Review · Reviewer_31bg · 2023-10-30

**Soundness:** 3 good
**Presentation:** 2 fair
**Contribution:** 2 fair
**Rating:** 3
**Confidence:** 4

**Summary:**

This paper proposes a model for the MutualFriend dialogue task: where two players each see their own private list of entities with attribute-value pairs, with one entity shared between the players, and must talk to each other to identify this common entity. The approach uses theory-of-mind modeling, which has models predict first- and second-order belief states of  (1) what entities a player has and (2) what entities the player thinks the other player has. The paper collects annotations for these belief states, and uses them as an auxiliary loss and intermediate representation in neural dialogue models.

**Strengths:**

S1) I found the topic of theory-of-mind in dialogue exciting, and timely with the recent success of LLMs for dialogue but their limitations in perspective taking and common ground. MutualFriends is a good dataset to use for this, given that it has natural language and interactions but a discrete and tractable set of entities to maintain beliefs over.

S2) The general idea of the approach seemed very reasonable. I appreciated the separate modeling of first- and second- order beliefs, and defining common ground using these. If I understood the model right, it uses the belief states only as an auxiliary loss, not as an input to the model.

S3) The evaluation was thorough, using a range of auxiliary tasks and evaluating three different model architectures, with ablations of each of the belief states. I also appreciated that the paper did a human evaluation! (But see below for some comments.)

**Weaknesses:**

W1) From the automatic evaluation generation scores (BLEU-2 < 5), self-play success rates being low (<10%, with very long games, > 20 turns in most cases), and few qualitative outputs given, none of the trained (GRU, Transformer, or BART) systems appear to be working very well. This made it difficult to draw conclusions from the results, and additionally I didn't see clear trends in the various settings.

W2) The writing of the paper was rough in parts, and I found the description of the modeling in particular hard to follow -- both in the low-level details, but also in the high-level intuitions for what belief states are over. See questions below.

W3) It would be helpful to evaluate the general approach on another common ground dataset. Some possible ones might be OneCommon [1,2] (which has the advantage of having annotated referents, which might be compatible with the belief states used here, as explored in [3]) or PhotoBook [4]. However, given the complex visual grounding in those two datasets, I don't think this is a crucial weakness.

[1] Udagawa and Aizawa, A Natural Language Corpus of Common Grounding under Continuous and Partially-Observable Context. AAAI 2019

[2] Udagawa and Aizawa, An Annotated Corpus of Reference Resolution for Interpreting Common Grounding. AAAI 2020

[3] Chiu et al. Symbolic Planning and Code Generation for Grounded Dialogue. EMNLP 2023]

[4] Haber et al. The PhotoBook Dataset: Building Common Ground through Visually-Grounded Dialogue. ACL 2019

**Questions:**

Q1) I was confused about what beliefs are over, and what sort of language should produce a change in e.g. b_A and b_{BinA}. Does b_A being high for an entity mean that A believes that they have that entity? Does b_{BinA} being high for an entity mean that A believes that B has that entity? How does this interact with what entities are mentioned in the dialogue utterances? I couldn't come up with an explanation that fully explained Fig 1, Fig 2, and the text in the paper.  (e.g. in Fig 2, B asks about Yo-yoing in Turn 1 and Drama in Turn 3, but the b_A dynamics annotations differ).

Q2) Do "entities" refer to values that the attributes can take on (e.g. Hobby = Drama), or full rows in the people tables (e.g. Drama, Diane, Morning)? It seems like the first one from the equations (1) and (2), but "entities" is a bit confusing as a name for this.

Q3) Are the 1 / 0 / -1 belief states balanced? It seemed surprising to me that a random guess is claimed to get 0.33 (F1?) in section 4.2

Q4) Can more details be given about the human evaluations? I was confused about how all of "grammar correctness, efficiency of finding the common friend" and presumably also task success could be evaluated in a single pairwise comparison score.


Other comments:
- The definition of common belief as "the gap between the speaker's belief and her belief estimation of the listener" was interesting but a bit counterintuitive to me. It would help to justify this more.
- The way b_m is implemented, it seems that later belief dynamics updates will have less of an effect on the belief state, as b_0 is initialized to zeros so having delta = 1 or -1 in each turn will have diminishing impact post-softmax. "-1 indicates the disappearance of an entity in the belief" was confusing, since the belief won't be zeroed out (post-softmax) unless -inf is added in, if I understand right.
- I found the model description in Section 3 pretty hard to follow. It would help to give a much clearer definition of the first- and second- order beliefs, the intuition for b_diff and how it's used, and then have the details of implementation (tanh, etc) be secondary to this. The Figure 2 is a nice step in this direction but I'd also recommend relying much more on the figure in the text.

---

> ### Author Response · Authors · 2023-11-22
>
> Thank you for your thorough suggestions and comments.
>
> > The performance is overall low, making it difficult to draw conclusions from the results.
>
> **A:** Since the entity mentioned in the next utterance can be pretty sparse, the common belief estimation can be the bottleneck for the performance improvement. We will explore this problem in the future work.
>
> > Model section hard to follow.
>
> **A:** We will rewrite section 3 in the revision.
>
>
> > What beliefs are over and what sort of language should produce a change.
>
> **A:** $b_A$ being high means that A believes that the mutual friend should have that value(entity) and the value should also be in the A's private knowledge list. $b_{BinA}$ being high means that A believes that B believes that the mutual friend should have that value(entity) and the value should also be in A's private knowledge list or in the context. When speakers mention some values(entities) in the utterances, the corresponding values(enitities) in the beliefs will be high. When the speakers negate the values(entities), the corresponse values(entites) will be -1. For example, in figure 2, when B mentions "yo-yoing", since there is "yo-yoing" in A's knowledge list, it will only influence $b_{BinA}$. $1$ when B mentions it and $-1$ when A negates it. For "Drama", it will influence both $b_A$ and $b_{BinA}$. $1$ when A mentions it.
>
> > What "entities" refere to?
>
> **A:** Yes, the "entities" refer to the values that the attributes can take on. We adopt the same naming conventions in the MutualFriend dataset paper.
>
> > Are belief dynamics balanced?
>
> **A:** No, most of the belief dynamics stay "0" as unchanged. We compute F1 as the macro-average over each class, therefore approxiamtely 1/3.
>
> > Details about "grammar correctness, efficiency of finding the common ground friend".
>
> **A:** Our user interface is provided in the Appendix. For "grammar correctness", we ask the subjects to compare "which sentence is more grammarly correct". For "efficiency", we are not asking whether the sentence can lead to the task success, we are asking based on the current speaker's knowledge list, "which sentence expresses the information more efficiently by faster converging the talk meanwhile mentioning fewer entities".
>
> > The definition of common belief was interesting but a bit conterintuitive.
>
> **A:** We define the common belief as the next most possible entities to be mentioned or the common ground they want to negotiate next.
>
> > It seems that later belief dynamics updates will have less of an effect on the belief state. "-1 indicates the disappearance of an entity in the belief" was confusing since the belief won't be zeroed out.
>
> **A:** The belief does not need to be zeros out. We only need to maintain a distribute over all entities about which the speakers focus more. We aggregate belief dynamics over turns to get the sense of the speakers' attention. Entities mentioned will be strengthened with "+1" and entities negated will be weakened with "-1".

---

> ### Comment · Reviewer_31bg · 2023-11-22
>
> Thanks to the authors for engaging in the response! The clarifications about the belief states are helpful, and I appreciate the other comments too.
>
> While I don't think the paper is ready for acceptance yet, I'd like to repeat that I am really excited about this direction of work, and look forward to hopefully seeing an improved version of this paper in the future.

---

### Official Review · Reviewer_bi42 · 2023-11-01

**Soundness:** 2 fair
**Presentation:** 2 fair
**Contribution:** 2 fair
**Rating:** 5
**Confidence:** 3

**Summary:**

The paper introduces a framework called MindDial, which incorporates Theory of Mind modeling, belief dynamics tracking, and response generation. For the framework, the authors annotated the MutualFriend dataset with information on belief dynamics. The framework comprises a knowledge encoder, context encoder, entity encoder, speaking act classifier, belief prediction module, and a response decoder, which are jointly trained with three objectives. They test three fine-tuned base models (GRU, transformer, BART) on tasks for mind prediction and response generation.

**Strengths:**

Incorporating different theory of mind modules for dialogues is interesting.

**Weaknesses:**

- The models that are used in the experiments need serious updates. Despite there are lots of pre-trained models that can be used off the shelf for dialogues, the authors are using GRU, vanilla transformer, and BART on a dialogue task. The readers will be curious whether the complicated modules actually do add up to real-world performance.
- Moreover, since the framework is on top of MutualFriend, a dialogue dataset with a very limited scope and responses that lacking naturalness, the framework’s generalizability is questionable.
- The performance improvement when the generator copies from the common belief distribution is also very small (Table 2), which casts doubt on the effectiveness of the proposed complicated methods.

**Questions:**

- Can the modules help existing models, such as other large language models (LLMs)? Or will LLMs perform much better without them?
- Have you given few-shot samples to models such as GPT-4?
- Why do you think the improvement is small in Table 2?

---

> ### Author Response · Authors · 2023-11-22
>
> Thank you for your thorough suggestions and comments.
>
> > The models that are used in the experiments need serious updates.
>
> **A:** We will update the model backbones in the future revision.
>
> > Since the framework is on top of MutualFriend, the framework's generalizability is questionable.
>
> **A:** We want to clarify that the framework is not designed specific to MutualFriend. The mind gap between the first and second-order belief can be estimated for general dialy dialogues. However, as reviewer 31bg mentioned, for evaluation purpose, MutualFriend is the only dataset with clearly defined belief states.
>
> > The performance improvement is very small.
>
> **A:** Since the entity mentioned in the next utterance can be pretty sparse, the common belief estimation can be the bottleneck for the performance improvement. We will explore this problem in the future work.
>
> > Can the modules help existing models, such as other large language models?
>
> **A:** Thanks for your suggestion. We will expand our experiments in the future revision.

---

> > ### Comment · Reviewer_bi42 · 2023-11-23
> >
> > Thank you for the response.
> > I believe the improved future version of this work can be stronger with additional analysis and experiments.

---

### Official Review · Reviewer_3axi · 2023-11-08

**Soundness:** 1 poor
**Presentation:** 1 poor
**Contribution:** 2 fair
**Rating:** 3
**Confidence:** 3

**Summary:**

The paper proposes a supervised approximation method for decentralized POMDPs. The approach, MindDial, uses a supervised model of belief dynamics for modeling the agent and partner's belief, which can then be intersected to obtain the common belief. This belief model is used in a dialogue system and evaluated on MutualFriends. Evaluation shows that supervised belief dynamics are accurate on MutualFriends, slightly improve response generation, and improve success rates in selftalk or selfplay settings.

**Strengths:**

I believe the belief dynamics prediction idea is original, and such an approach would make theory of mind modeling computationally efficient.
This is a significant problem in multi-agent collaboration.
The biggest issue in this paper is that the belief dynamics are fully supervised. Prior work uses Bayes' rule to avoid specifying belief dynamics directly, instead opting to use models of observations given state in order to move the belief toward states that are consistent with observations.

**Weaknesses:**

The writing and clarity of the paper are poor. The number of unique symbols in section 3 is unnecessarily high due to poor choices in abstraction. Symbols within section 3 do not align with usage in other parts of the model. For example, I do not know how $d_t^{KB}$ is constructed. Presumedly, it is constructed from $d_A$, $d_{BinA}$, and $d_{bdiff}$, but I am not sure. I recommend rewriting section 3 to describe the model at the level of conditional probability distributions, rather than mixing parameterizations in as well. The only part of the model that could acceptably be described below the level of conditional probability distributions would be the belief dynamics. However, the belief dynamics could likely be described in terms of Bayes' rule as well. I give more detailed feedback in the questions section.

Another weakness is the experimental setup. The most important evaluation in task-oriented dialogue is *full* dialogue success when playing the *full* game with human partners. This is missing in the paper. The second most important evaluation is selfplay, which is a cheaper approximation of human evaluation. I believe this is presented in Table 4, but surprisingly the success rates are all around or below 10%. It is unclear what this number represents, as it could be reporting the success rate per game, per selection, or even per turn. Additionally, the original MutualFriends paper [1] had success rates greater than 76% for full games, and greater than 18% per selection.

Additionally, MutualFriends has very simple belief dynamics. If a property of an entity is mentioned, that is positive evidence for the entity until the entity is rejected. There is little uncertainty regarding interpretation and no gradation of belief updates. This makes annotations easy, but also brings into question whether such an annotation scheme is feasible for a realistic dialogue task. This would also certainly affect human evaluation of belief dynamics, which would be subjective and therefore have high variance.

**Questions:**

## Questions and comments
* The transactional and constructivist models, if borrowing those terms from another field such as psychology, should have citations. If not, why introduce them? They are simply either ToM or not, and first-order beliefs or second-order beliefs.
* Where does the idea of "five mind representation" appear in the paper? Figure 2 only shows 2 belief distributions, and their intersection (the common belief). The contributions only list 2 belief distributions and their intersection as well.
* What is the difference between $w_t$ and $y_t$? How come equation (5) uses $w_t$ but equation (6) uses $y_t^l$?
* The superscript $m$ is inconsistent within the belief prediction subsection of section 3, appearing in equations (1) and (2) but absent at the bottom of page 4.

## Writing suggestions
Clarity comes from economical choices in abstractions. This means focusing the story on the aspects (models, variables) that convey your contribution. Linearly describing every aspect of the model means that any gaps in the coverage (see questions above) will be even more confusing. Instead, describe what you need to convey the contributions in a top-down manner. Here are a couple of examples.
1. *Model abstractions* Section 3 should focus primarily on the belief dynamics model, which is the contribution of the paper. There are currently 4 models: Utterance encoder, Belief dynamics, Dialogue act, and Response decoder. I suggest only having two models: belief dynamics and response decoder, and abstracting away the other models into the parameterization of these two main models. Additionally, the response decoder is not part of the contribution, and should not be heavily focused on. The low-level parameterization details should be present in the paper but can be relegated to the appendix.
2. *Variables*: The most important aspects of the belief dynamics model are predicting the increments given an utterance, aggregating increments, and comparing two belief distributions. The goal here is to minimize the number of subscripts and superscripts.

## Suggested citations
1. Please add, at the very least [1]: A decentralized POMDP approach for the Cards corpus. Your approach's difference: Directly models belief dynamics, which needs annotation but is also much more computationally efficient.
2. More citations can be found in [2]

[1] Adam Vogel, Max Bodoia, Christopher Potts, and Daniel Jurafsky. 2013. Emergence of Gricean maxims from multi-agent decision theory. In Proceedings of the Conference of the North American Chapter of the Association for Computational Linguistics (NAACL), pages 1072–1081, Atlanta, Georgia. Association for Computational Linguistics

[2] Fried, D., Tomlin, N., Hu, J., Patel, R., & Nematzadeh, A. (2022). Pragmatics in Language Grounding: Phenomena, Tasks, and Modeling Approaches.

---

> ### Author Response · Authors · 2023-11-22
>
> Thank you for your thorough suggestions and comments.
>
> > The writing and calrity of the paper are poor.
>
> **A:** Thanks for your thorough suggestion. We will rewrite section 3 accordingly.
>
> > How $d_t^{KB}$ is constructed.
>
> **A:** It is shown in Equation 3, where the decoder will attend to the output sequence of the knowledge encoder to get the attention score. Then the knowledge representation is contructed by aggregating the output sequence weighted by the attention score.
>
> > Full game playing with human partners missing
>
> **A:** Thanks for your suggestion. We will expand our experiments in future revision.
>
> > The meaning of success rate in Table 4 is not clear. The success rate is low compared with the original paper.
>
> **A:** We report the success rate per game. The low success rate could be attributed to the absence of structured knowledge and logical rules in the model, elements which might improve efficiency in this task.
>
> > MutualFriends has very simple belief dynamics. It brings into question whether such an annotation scheme is feasible for a realistic dialogue task.
>
> **A:** In this work, We incorporate belief annotation only for quantative evaluation purpose. The subjectivity and open-domain nature of beliefs definitely makes the data collection and annotation a challenging task. More comprehensive belief dynamics collection and annotation scheme will be considered in the future work.
>
> > Transactional and constructivist model citation missing
>
> **A:** We will add citations correspondingly.
>
> > Where does the idea of "five mind representation" appear in the paper?
>
> **A:** It is introduced in Introduction and Figure 1. Since we have two agents, we model two first-order, two second-order mind and a third-order common mind.
>
> > What is the difference between $w_t$ and $y_t$?
>
> **A:** $w_t$ is one word in $y_t$. Since we model the conversation as free-form, there can be consecutive responses $\{y_t^1,..., y_t^L\}$ from one speaker at each turn.
>
> > The superscript $m$ missing at the bottom of page 4.
>
> **A:** Thanks for pointing it out. We will modify it in the revision.